# Fetal Arrhythmia Detection Based on Labeling Considering Heartbeat Interval

**DOI:** 10.3390/bioengineering10010048

**Published:** 2022-12-30

**Authors:** Sara Nakatani, Kohei Yamamoto, Tomoaki Ohtsuki

**Affiliations:** 1Graduate School of Science and Technology, Keio University, Yokohama 223-8522, Kanagawa, Japan; 2Department of Information and Computer Science, Keio University, Yokohama 223-8522, Kanagawa, Japan

**Keywords:** fetal electrocardiogram, fetal arrhythmia, deep learning

## Abstract

Arrhythmia is one of the causes of sudden infant death, and it is very important to detect fetal arrhythmia for fetal well-being. Fetal electrocardiogram (FECG) is one of the methods to detect a heartbeat. Fetal arrhythmia can be detected based on the heartbeat detection results from FECG signals such as heartbeat intervals. However, the accuracy of arrhythmia detection easily degrades depending on the accuracy of heartbeat detection. In this paper, we propose a deep learning-based fetal arrhythmia detection method using FECG signals. Recently, arrhythmia detection methods using adult ECG signals have achieved a high arrhythmia detection accuracy based on deep learning. Motivated by this fact, in the proposed method, the acquired FECG signals are segmented, and the segments are input into a deep learning model that classifies them into normal or arrhythmia ones. Based on the classification results of multiple segments, a subject is judged as a healthy or arrhythmia subject. Each segment of the training data is divided into three categories based on the estimated heartbeat interval: (i) normal, (ii) arrhythmia, and (iii) a segment that could be both normal and arrhythmic. Only segments labeled as normal or arrhythmia are used for training a deep learning model to achieve a higher classification accuracy of the model. Through these procedures, the proposed method detects fetal arrhythmia with fewer effects of heartbeat detection results. The experimental results show that the proposed method achieves 96.2% accuracy, 100% specificity, and 100% recall, improving the values of conventional methods based on heartbeat detection and feature detection.

## 1. Introduction

Fetal heart rate (FHR) is an important indicator for accessing fetal health status. The risk of unexplained, unexpected fetal death is particularly high during the prenatal and perinatal periods. Fetal arrhythmia is one of the causes of sudden infant death [1,2]. Fetal arrhythmia is an abnormality of the fetal heartbeat, and is generally diagnosed as tachycardia if it is greater than 180 beats per minute and bradycardia if it is less than 100 beats per minute [3]. Fetal arrhythmias can be treated with antiarrhythmic drugs if diagnosed promptly [4]. Therefore, there has been great interest in the efficient diagnosis of fetal arrhythmia, and various fetal arrhythmia detection methods have been investigated. Ultrasonography, echocardiography, electrocardiography, and magnetocardiography are methods to collect fetal biological signals for this purpose [5]. Among these methods, fetal electrocardiogram (FECG) acquisition can be divided into invasive and non-invasive types. The invasive type involves a large physical burden on the mother and fetus [6]. In contrast, a non-invasive fetal electrocardiogram (FECG) can be used to obtain information on the electrical activity of the fetal heart from electrodes attached to the maternal abdomen [7]. Thanks to the low price of the used device, FECG has been widely used to detect the fetal heartbeat. The most conventional fetal arrhythmia detection methods are based on fetal heartbeat detection [8,9,10]. Specifically, in the conventional methods [8,10], the heartbeat is firstly detected, and then features are extracted from FECG signals based on the heartbeat detection results. Using the features, a subject is classified into healthy or arrhythmia. In addition, the conventional method [9] estimates FHR, extracts features from the variability of the FHR and classifies a subject into healthy or arrhythmia. The arrhythmia detection accuracy of these conventional methods depends on the accuracy of heartbeat detection. However, there is a problem that the fetal ECG signal is easily distorted by external disturbances such as maternal heartbeats and fetal body movements, and the accuracy of heartbeat detection is easily degraded [11]. In other words, as the accuracy of heartbeat detection deteriorates, the accuracy of fetal arrhythmia detection also deteriorates. Hori et al. have proposed a method for arrhythmia detection based on deep learning using adult ECG signals on data from 48 adult arrhythmia patients in the MIT-BIH arrhythmia database [12,13]. Since ECG patterns vary from subject to subject, developing an arrhythmia detection method that can accommodate different waveform patterns will enable the detection of abnormalities that could not be detected by the conventional methods. In this method, a versatile ECG abnormality determination method is proposed using an autoencoder and a convolutional neural network (CNN). Specifically, the autoencoder learns only normal waveforms that are easy to collect and acquires features of normal waveforms. The CNN is then used to compare the features obtained from the ECG waveforms targeted for arrhythmia detection with the features of the normal waveforms of healthy subjects. The results are used to determine whether the input ECG waveform is a normal waveform or an abnormal one containing arrhythmia based on a threshold value. This makes it possible to take into account the differences in waveforms for each subject. Gyohten et al. have also proposed a method to detect any arrhythmia by building a normal ECG model using deep learning in 42 adult arrhythmia patients in the MIT-BIH arrhythmia database [12,14]. This method uses CNN and long short-term memory (LSTM) to build a model of the normal ECG signal. This model takes a normal ECG signal as input and learns to predict subsequent normal ECG signals. When an abnormal ECG signal is input, the model can predict the subsequent ECG signal, which is far from the actual ECG signal, and can determine whether the input signal is abnormal or not. This means that the method can determine any arrhythmia because it does not require prior knowledge of annotation. Next, we present other literature that applies deep learning models to FECG signals. In [15], a multichannel signal quality classifier for FECG waveforms is presented. Each recording was labeled as a whole and assigned a quality class label of good or bad. Preference was given to recordings with consistent perceived signal quality to ensure that the entire recording was labeled correctly. Recordings with inconsistent signal conditions were discarded. Labels were assigned by the study’s data engineers based on FECG visibility and perceived SNR of the abdominal channel (hospital clinicians were consulted for the decision process). A deep learning model was developed in the literature [16] to automatically identify maternal heart rate (MHR) and more generally false signals (FS) in FHR recordings. The model can be used to preprocess FHR data prior to automated analysis or as a clinical alert system to assist practitioners. The training data set included data annotated by experts on the observed records. These two references also use deep learning models to classify FECG signals. It is important to note, however, that these two references are concerned with signal accuracy. Arrhythmia is not only a disturbance of the signal waveform, but also includes abnormalities in the heartbeat interval. Therefore, arrhythmias may not be correctly classified as arrhythmias using these methods [15,16].

In this paper, we propose a fetal arrhythmia detection based on deep learning using FECG signals. A fetal arrhythmia detection method based on deep learning has been proposed in our paper [17]. In the proposed method, the acquired FECG signals are first segmented to include several heartbeats. The FECG segment is fed into a deep learning model that classifies each segment as normal or arrhythmia. The classification results of the model are used to determine whether each subject is a normal or arrhythmia subject. Each segment of training data is labeled as a normal segment, an arrhythmia segment, or a moderate segment that may be associated with both normal and arrhythmia ones, based on the estimated heartbeat intervals. To improve the classification accuracy of the model, only normal and arrhythmia segments are used for training; moderate segments are not used. This deep learning-based arrhythmia detection is expected to prevent arrhythmia detection accuracy from being degraded by errors in heartbeat detection, since the ECG signal waveform itself is segmented as input data for the model. To evaluate the performance of our proposed method, we carried out the performance evaluation based on the dataset containing 12 arrhythmia subjects and 14 normal subjects [18]. As a result, the proposed method achieved at most 96.2% of accuracy, 100% of recall, and 100% of specificity in the binary classification of healthy and arrhythmia subjects. In addition, the experimental results showed that our method could outperform the other existing methods [8,9,10] in arrhythmia detection accuracy. Here, it is worth mentioning that there are two major differences between this paper and [17]. First, this paper describes the parameter setting and segment labeling methods in the proposed method and clarifies the rationale for these methods through more detailed evaluation and discussion. Specifically, we explain the impact of different parameter settings on the proposed method and provide the rationale for the parameter settings. We also give the results of characterization using additional metrics for different parameter settings and emphasize the justification for the values we set. Furthermore, a more detailed discussion of the experimental results will describe specific issues with the method and possible improvements. Second, this paper contains a new comparison and discussion with the characterization results of a new method that takes into account the problems of the proposed method. Specifically, the segments are labeled taking into account the differences in the mean and standard deviation of the heartbeat intervals for each subject. The subjects are then binarized into healthy subjects and arrhythmia subjects based on the results of the three-value classification of the segments.

The rest of this paper is organized as follows: In Section 2, we describe related research on fetal arrhythmia detection. In Section 3, we explain the proposed method. In Section 4, we evaluate the performance of our method. Finally, we conclude this paper in Section 5.

## 2. Related Work

In this section, we describe previous research related to fetal arrhythmia detection via FECG. FECG can be derived from AECG [19,20]. To detect fetal heartbeat based on abdominal ECG (AECG), the conventional method [19] detects the maternal QRS wave from the AECG by using the Pan–Tompkins algorithm [21], creates a template of the maternal ECG (MECG) based on the detected QRS wave, and subtracts the template from the AECG to extract the FECG. FHR is estimated by detecting peaks due to fetal heartbeat over the extracted FECG. However, the extracted FECG contains not only fetal heartbeat components but also noise that could cause wrong peak detection. To improve the accuracy of FHR extraction, Niida et al. have proposed a fetal heart rate estimation method using the first derivative of the FECG signal and multiple weighting functions [20]. In this method, the first derivative of the FECG signal is calculated to acquire R-peak candidates. To emphasize the actual R-peaks, the amplitudes of the peaks are weighted based on the assumption that the RR intervals follow a Gaussian distribution.

With these FHR estimation methods, some researchers have proposed fetal arrhythmia detection [8,9,10]. All of these conventional methods use PhysioNet’s public database [18]. The conventional method [8] firstly detects the characteristic points of the FECG signal, namely Q, R, S, and T points, and calculates the average values of RR, SS, QQ, ST, and TT intervals. Furthermore, the QRS interval reflects the electrical activity of the heart during ventricular contraction and is the most important segment of the ECG signal [22]. Therefore, the average amplitude and width of the QRS interval for each ECG are calculated and used as features as well. In addition to that, the nonlinear operator of energy tracking Teager Energy Operator (TEO), which includes the nonlinear behavior of the RR interval, is also extracted as a feature [23]. Using these features, a support vector machine (SVM) classifier is trained to classify a subject as healthy or arrhythmia. The Leave One Out (LOO) cross-validation method is used to evaluate the learning model. The experimental results have shown that this method achieves a specificity of 91.7%, a recall of 75.0%, and an accuracy of 83.3%. In the conventional method [9], independent component analysis (ICA) and singular value decomposition (SVD) are used to estimate the FHR by removing the maternal ECG components, and the FHR is segmented. Next, the entropy-based features, which represent the similarity between adjacent segments, are calculated from the FHR variation. A threshold is set for the entropy-based features, and a subject is classified as healthy or arrhythmia. A specificity and a recall of this method are 100% and 91.0%, respectively. The conventional method [10] also uses ICA to remove the maternal ECG component to obtain the FECG. Features are then extracted using a peak detection algorithm to identify the extracted FECG signal. Specifically, the peak detection algorithm using the state machine logic is used to extract features such as the RR interval, ST interval, the width of the QRS wave, and the amplitudes of QRS and T waves from the FECG. Binary classification of the subjects with healthy and arrhythmia is performed using a naive Bayes classifier. The naive Bayes classifier has the advantage of requiring fewer inputs of feature values. Through the experiments, this method has been shown to achieve a specificity of 96.3%, a recall of 74.8%, and an accuracy of 93.7%. However, the fetal heartbeat components can be easily distorted by maternal heartbeat components and fetal body motion, which tends to degrade the accuracy of fetal heartbeat detection. Nevertheless, these conventional fetal arrhythmia methods [8,9,10] are based on fetal heartbeat detection from the FECG, and the accuracy of arrhythmia detection depends on the heartbeat detection accuracy. Therefore, it is necessary to develop a fetal arrhythmia detection method that is less dependent on fetal heartbeat detection.

## 3. Proposed Method

In this section, we explain the proposed fetal arrhythmia detection using deep learning.

### 3.1. Framework of Proposed Arrhythmia Detection

First, FECG is extracted from AECG, and the RRI (RR-Interval) is estimated using the fetal heartbeat detection method [20]. In this method, since the output of the A/D converter is contaminated by fetal body motion, the preprocessing is performed to remove saturated data and invalid data from the raw AECG. Next, the maternal R-peaks are extracted from the preprocessed AECG. After the locations of maternal R-peaks are detected, the maternal cycles are removed from the AECG, and FECG is extracted. In addition, to detect the fetal heart rate, the candidates of the R-peak are generated. By designing a weighting function, we can detect the fetal R-peaks from the candidates. Figure 1 and Figure 2 show examples of the AECG and the extracted FECG, respectively. The acquired FECG signal is then down-sampled to 500 Hz. The FECG signal is then segmented by a 3 s-time window that contains several heartbeats. In addition, the step size of the window is set as 1 s.

The segments of FECG are classified as normal or arrhythmia segments based on deep learning. Here, note that it is still challenging to detect arrhythmia by using only RRIs, since the RRIs have some estimation errors. Therefore, for more accurate arrhythmia detection, we utilize the deep learning technique that performs the classification using the FECG waveform itself as input data. As a deep learning model, we use CNN which has been successfully applied for the classification task [24,25]. Figure 3 shows the structure of the CNN model used in the proposed method. The convolution layer with the activation function ReLU highlights the features of the input FECG signal, and then the pooling layer reduces the feature dimension. The combination of the convolution and pooling layers is repeated twice, which is followed by the affine layer. Finally, the input FECG segment is classified into normal or arrhythmia ones.

After the classification, a subject is judged based on the multiple results of the classification. Specifically, a threshold th is set for the ratio of the segments classified as arrhythmia by the CNN. A subject is judged as an arrhythmia one, when the ratio is greater than th; otherwise, a subject is judged as a healthy one.

### 3.2. Training Dataset

Arrhythmia is mainly determined by abnormal heartbeat intervals in ECG signals. However, since arrhythmia does not always occur, normal waveforms are considered to be included in the ECG signal data of arrhythmia subjects. Therefore, to train the CNN, the training data are labeled based on the estimated RRI. Figure 4 shows the distributions of the estimated RRIs for all the arrhythmia and normal subjects in the database used in this study. As can be seen from Figure 4, the distribution of the estimated RRIs of the healthy subjects is clustered around 400 ms, whereas those of the arrhythmia subjects are relatively widely distributed. The standard deviations of the respective estimated RRIs are 33.94 ms for the healthy subjects and 71.73 ms for the arrhythmia subjects. In the proposed method, the estimated RRIs are classified into three categories, i.e., (i) normal, (ii) moderate, and (iii) arrhythmic RRIs, by a 25 ms-RRI range as shown in Figure 4. When the RRI range has more RRIs of healthy subjects than those of arrhythmia subjects, the RRIs within the RRI ranges are defined as normal RRIs. When the ratio of the number of healthy subjects’ RRI to that of the arrhythmia one exceeds *x*, the RRIs within the RRI ranges are defined as moderate RRIs. Here, it is worth mentioning that moderate RRIs might be associated with both healthy and arrhythmia subjects. When such a ratio is lower than or equal to *x*, the RRIs within the RRI ranges are defined as arrhythmic RRIs.

The FECG segments are then labeled based on the three types of RRIs. Specifically, each segment is labeled as a normal segment, when the segment contains only normal RRIs. In addition, each segment is labeled as a moderate segment, when the segment contains at least one moderate RRI without arrhythmic RRIs. Each segment is labeled as an arrhythmia segment, when the segment contains at least one arrhythmic RRIs. Figure 5 shows examples of the three types of the FECG segments. To improve the classification accuracy of the CNN, moderate segments that could be associated with both normal or arrhythmic RRIs are not used to train the CNN model, meaning that only normal and arrhythmia segments are used for the training. The arrhythmia segment generated from a normal subject is then treated as a moderate segment.

## 4. Performance Evaluation

### Evaluation Setup

In this study, we used the AECG database containing fetal arrhythmias provided on the PhysioNet website [18]. Table 1 lists the experimental specification. Each of the subjects’ AECG recordings consists of four or five channels, and one maternal thoracic channel. The database contains about 10 min of AECG signals from 12 arrhythmia and 14 healthy subjects. The sampling frequency of the AECG signal is 500 Hz or 1 kHz. The data set is also labeled as healthy or arrhythmic for each subject. The loss function for the network is binary cross-entropy, and the optimizer is Adam with lr=0.0003. The model was implemented using Keras v1.1.2 on TensorFlow v2.4.0.

To demonstrate the accuracy of the proposed method in detecting fetal arrhythmia, we evaluated the binary classification accuracy of healthy and arrhythmia subjects to the proposed and conventional methods [8,9,10]. In the conventional method [8], the subject is classified using SVM based on features extracted from the FECG. In the conventional method [9], the subject is classified by using threshold values based on the entropy features extracted from fetal heart rate variability. In the conventional method [10], the features are extracted from FECG based on the peak detection algorithm using the state machine logic, and the subject is classified by using a naive Bayes classifier. As the performance metrics, we calculated a specificity, a recall, and an accuracy using the following equations:(1)$$specificity=\frac{TN}{TN+FP},$$
(2)$$recall=\frac{TP}{TP+FN},$$
(3)$$accuracy=\frac{TP+TN}{TP+FP+TN+FN},$$
where TN represents the number of healthy subjects that are correctly classified as healthy subjects. FN represents that of arrhythmia subjects that are incorrectly classified as healthy subjects. FP represents that of healthy subjects that are incorrectly classified as arrhythmia subjects. TP represents that of arrhythmia subjects that are correctly classified as arrhythmia subjects. As the experimental parameters, the threshold *x* for RRI classification was set as 0.63 based on the results of the search, and the threshold th for the ratio of arrhythmia segments was varied from 0.00 to 1.00 in increments of 0.05.

## 5. Results

Table 2 shows the ratio of the number of RRIs of healthy subjects to that of RRIs of arrhythmic subjects in Figure 4, and the classification results of each RRI range when each value of the threshold *x* for RRI classification is applied to it. Each column of Table 2 is described below. “RRIrange” indicates the value of RRI representing each range in Figure 4. “RRIratio” indicates the ratio of the number of healthy subjects’ RRI to that of arrhythmia one in each range in Figure 4. “The classification result” indicates the classification result for each range when the threshold x for the RRIratio is varied. “A” represents arrhythmic RRI, “M” represents moderate RRI, and “N” represents normal RRI. When the RRIratio is lower than or equal to *x*, the RRIs within the RRI ranges are defined as arrhythmic RRIs. When the RRIratio exceeds *x*, the RRIs within the RRI ranges are defined as moderate RRIs. When the RRI range has more RRIs of healthy subjects than those of arrhythmia subjects, the RRIs within the RRI ranges are defined as normal RRIs. The comparison is made for x=0.50,0.60,0.63,0.65,0.70. The values of *x* are set so that the three classification groups of arrhythmic RRI, moderate RRI, and normal RRI, which are assigned to each range of the histogram shown in Figure 4, are different from each other. From Table 2, it can be seen that the number of RRI ranges classified as moderate RRIs and arrhythmic RRIs changes when the threshold *x* is varied. The more RRIs classified as Moderate RRIs, the more FECG segments are labeled as moderate segments and the fewer segments are used to train the deep learning model. Figure 6, Figure 7 and Figure 8 respectively show the recall, the specificity, and the accuracy of the subject classification when segments are labeled by varying the threshold *x* used for RRI classification. A subject is judged as an arrhythmia one, when the ratio is greater than th; otherwise, a subject is judged as a healthy one.

Figure 6 shows that, for x=0.63,0.65,0.70, th≤0.75 gives the highest recall compared to other threshold *x*. The lower the threshold *x* used for RRI classification, the more RRIs are classified as moderate RRIs. Correspondingly, the more segments are tentatively labeled as moderate segments and the number of arrhythmia segments is reduced. Based on this, the number of segments used to train and test the deep learning model is reduced, and segments needed for subject classification may also be excluded. Therefore, the accuracy of arrhythmia subject classification is considered to have deteriorated. Figure 7 shows that the highest specificity is obtained at x=0.63 for almost all the thresholds th. When the threshold *x* used for RRI classification is high, the number of RRI data classified as moderate RRI decreases and that of RRI data classified as arrhythmic RRI increases. Based on this, the number of moderate segments would decrease and segments that could be both normal and arrhythmia could be used as arrhythmia segments. Therefore, the classification accuracy of the normal segments would be degraded, resulting in a degradation of the classification results for normal subjects. From Figure 8, it can be seen that, at x=0.60,0.63,0.65,0.70, 0.6≤th≤0.7, the highest accuracy is obtained compared to other threshold values of *x*. It can also be seen that, for x=0.63, high accuracy is obtained in the case of th≤0.35 compared to the other *x* values. This may be due to the same reason as for recall and specificity. Based on these results, x=0.63 was considered optimal for the proposed method for the data set used in this study.

Figure 9, Figure 10 and Figure 11 respectively show the recall, the specificity, and the accuracy of the proposed method for the threshold x=0.63 used for RRI classification and conventional methods [8,9,10]. In addition, for a better comparison, these figures include the result of our method with the simple labeling algorithm: the FECG segments of healthy and arrhythmia subjects are labeled as normal and arrhythmia ones, respectively. First, we compare the recall of the proposed method with those of the other methods.

From Figure 9, it can be seen that the proposed method outperforms the conventional ones [8,9] and provides almost the same recall as the conventional ones [10] in the range where 0.40≤th≤0.70. In these conventional methods, the arrhythmia subject is detected based on heart rate estimation and feature detection of the FECG signal. In contrast, our method classifies healthy and arrhythmia subjects based on the FECG signal waveforms using deep learning. Therefore, our method reduces the effect of errors induced by the heart rate estimation and the feature detection. In addition, our method achieves high recall, compared to the conventional one [10] in the range where th≤0.40. This is because the arrhythmia is determined when the ratio of the segments classified as arrhythmia exceeds the threshold th, and thus the smaller th makes it easier to judge that the subject has arrhythmia. Furthermore, our proposed method achieves a higher recall than our method with the simple labeling algorithm, which is brought by labeling the training data based on the RRI distribution.

In terms of specificity, as can be seen from Figure 10, the proposed method improves the specificity of the conventional ones [8,9], and achieves 100% specificity in the range where th≥0.6. As aforementioned, these results are brought by the arrhythmia detection that does not depend on heart rate estimation and the feature detection of the FECG signal. In addition, the larger th makes it easier to determine that the subject is healthy, since the subject is judged as normal when the ratio of the segments classified as arrhythmia is lower than the threshold th. In addition, as well as the result of the recall, we can see that our proposed method based on the RRI distribution outperforms our method with the simple labeling algorithm that labels all segments of arrhythmia subjects as arrhythmia segments and all segments of healthy subjects as normal segments. Therefore, it can be said that labeling segments based on the RRI resulted in higher classification accuracy for both normal and arrhythmia subjects. This may be because the effect of the estimation error of the RRI was reduced.

Finally, from Figure 11, it can be seen that the proposed method improves the accuracy of the conventional methods [8,9] by 16.7% and 2.44%, respectively, in the range where 0.6≤th≤0.7. Furthermore, our method achieves a higher accuracy than our method with the simple labeling algorithm for almost all the threshold values. In our method with the simple labeling algorithm, all the segments of arrhythmia subjects are labeled as arrhythmia one. However, arrhythmia does not necessarily happen all the time, meaning that some segments of arrhythmia subjects are not arrhythmia ones but normal ones. In contrast, our proposed method performs segment labeling based on the RRI distribution, and does not use the segments that might be the normal and arrhythmia ones. Therefore, our proposed method provides a high classification accuracy of the CNN, compared to our method with the simple labeling algorithm. Note that the proposed method requires threshold setting.

We focus on the classification results for the segments of healthy subjects. Table 3 shows the number of arrhythmia segments and normal segments when the segments of three healthy subjects in the database are labeled based on the estimated RRI and classified using CNN, and the ratio of segments classified as arrhythmia out of all segments. From Table 3, it can be seen that even segments from healthy subjects can be misclassified as arrhythmia segments. Therefore, if the threshold th used for subject classification for this percentage of arrhythmia segments is too low, the accuracy and the specificity deteriorate.

Based on this fact, in our future work, it is necessary to make our method robust to the threshold setting. In particular, since the lower th makes it easier to detect arrhythmia subjects, we will improve the accuracy for lower th.

Next, we consider how to take into account the differences of the mean and standard deviation of the RRI among subjects in order to improve the accuracy of a subject classification and increase versatility. Figure 12 shows a scatter plot of the mean and standard deviation of the RRI for each subject. From Figure 12, it seems that the mean and standard deviation of RRI vary widely among normal subjects. In the proposed method, RRIs were classified and segments were labeled for all normal subjects and all arrhythmia subjects collectively based on the histogram shown in Figure 4. In this method, the differences in the mean and standard deviation of RRIs for each subject could not be taken into account. Therefore, we will compare the proposed method with the method that takes into account differences in the mean and standard deviation of RRIs per subject in the following.

### 5.1. Method Based on 3-Value Classification Considering Differences in Mean and Standard Deviation of RRI

Each arrhythmia subject has a different mean and standard deviation of RRIs. The proposed method of labeling based on the RRI distributions of arrhythmia and healthy subjects compared above does not take this into account. Therefore, we describe a method that takes into account the differences in the mean and standard deviation of the RRI for each subject.

#### 5.1.1. Segment Labeling

Based on the mean and standard deviation of the RRI for each subject, the RRI is classified according to the following definition for arrhythmia subjects:(4)$$\begin{array}{c} moderate\_min=Mean\_RRI-y\times RRI\_std\\ moderate\_max=Mean\_RRI+y\times RRI\_std \end{array}$$
RRIs between moderate_min and moderate_max are defined as moderate RRIs; all others are defined as arrhythmic RRIs. Mean_RRI is the mean value of RRI for each subject, and RRI_std is the standard deviation of RRI for each subject. The percentage of data classified as moderate RRI among the RRI data changes depending on the value of *y*. Specifically, the larger *y* is, the greater the percentage of RRI data are classified as moderate RRI, and the smaller *y* is, the smaller the percentage becomes. The value of *y* is set so that the percentage of data classified as medium RRI is 50, 60, 70, 80, or 90%. After classifying the RRI data as moderate RRI or arrhythmic RRI for each arrhythmia subject based on Equation (Equation 4), we label the segments using the same definition as in the proposed method. Specifically, each segment is labeled as a moderate segment, when the segment contains at least one moderate RRI without arrhythmic RRIs. Each segment is labeled as an arrhythmia segment, when the segment contains at least one arrhythmic RRIs. Label all segments of normal subjects as normal segments.

#### 5.1.2. Classification of the Subject

Based on the results of the three-value classification of the segments, the subjects are binary classified as either arrhythmia subjects or normal subjects. The segments of arrhythmia subjects are labeled considering the mean and standard deviation of the RRI for each subject. All segments of normal subjects are labeled as normal segments. These segments are classified into three values, arrhythmia segment, medium segment, and normal segment, using a deep learning model as in the proposed method. The subject is then classified based on the ratio of arrhythmia segments to the total number of normal and arrhythmia segments among the classification results. This ratio is calculated based on the following equation:(5)$$ARR\_NR\_ratio=\frac{ARR\_segments}{ARR\_segments+NR\_segments}$$
ARR_segments represents the number of segments classified as arrhythmia, and NR_segments represents the number of segments classified as normal. ARR_NR_ratio represents the ratio of arrhythmia segments to the total number of normal and arrhythmia segments. For this ARR_NR_ratio, a threshold th is set as in the proposed method: if ARR_NR_ratio is greater than th, the subject is determined to be arrhythmic; otherwise, the subject is determined to be normal.

#### 5.1.3. Comparison of the Proposed Method with the Method Based on Three-Value Classification

Figure 13, Figure 14 and Figure 15 show the recall, specificity, and accuracy calculated by the method based on the three-value classification and the proposed method, respectively. However, in the method based on the three-value classification, the percentage of RRI data classified as moderate RRI is varied to 50, 60, 70, 80, and 90%. Figure 13 shows that the proposed method achieves the highest recall at almost all threshold th. Comparing the methods based on three-value classification, the highest recall is achieved when the percentage of classification to moderate RRI is 50%. However, looking at the overall value trends, it is not necessarily the case that the recall improves with a lower percentage of classification to moderate RRI. Next, Figure 14 shows that the method based on three-value classification achieves greater specificity than the proposed method for 0.2≤th≤0.55, when 90% of the methods are classified as moderate RRI. Similarly, when the ratio is 80%, the method based on three-value classification achieves greater specificity than the proposed method when 0.35≤th≤0.55. Consider this: the greater the percentage of data classified as moderate RRI, the fewer the number of data classified as arrhythmic RRI. In other words, fewer segments are labeled as arrhythmia segments. This is expected to reduce the number of segments learned as arrhythmia segments in the deep learning model, and therefore the number of segments classified as arrhythmia segments will also decrease. Therefore, it is believed that a subject is more likely to be determined to be a normal subject when the percentage of segments classified as moderate RRI is increased. Finally, Figure 15 shows that the proposed method achieves the highest accuracy at almost all thresholds. It also seems that the highest accuracy is achieved for 0.35≤th≤0.4 when 50% of the methods are classified as moderate RRI among the methods based on three-value classification. However, looking at the overall trend of values, as in the discussion for recall, it can be considered that specificity does not necessarily improve with a lower percentage of classification to moderate RRI. This result suggests that there is no significant relationship between the percentage of subjects classified as moderate RRI and the accuracy of their classification.

Based on the above results, we believe that the proposed method is the most capable of correctly classifying subjects.

## 6. Conclusions

In this paper, we proposed the fetal arrhythmia detection method based on deep learning using FECG signals. The proposed method uses the CNN that classifies FECG signal segments as normal or arrhythmia ones. Since the classification is based on the FECG signal itself, it could be possible to reduce the effects of errors in FHR estimation and feature detection. In addition, by removing the FECG signal segment that might be associated with both the normal and arrhythmia ones, our proposed method improves the classification accuracy of the CNN. The experimental results show that the proposed method achieves 96.2% accuracy, 100% specificity, and 100% recall, improving the values of conventional methods based on heartbeat detection and feature detection. In our future work, we will consider improving the sensitivity to the threshold used in our method for more practical arrhythmia detection.

## Figures and Tables

**Figure 1 bioengineering-10-00048-f001:**
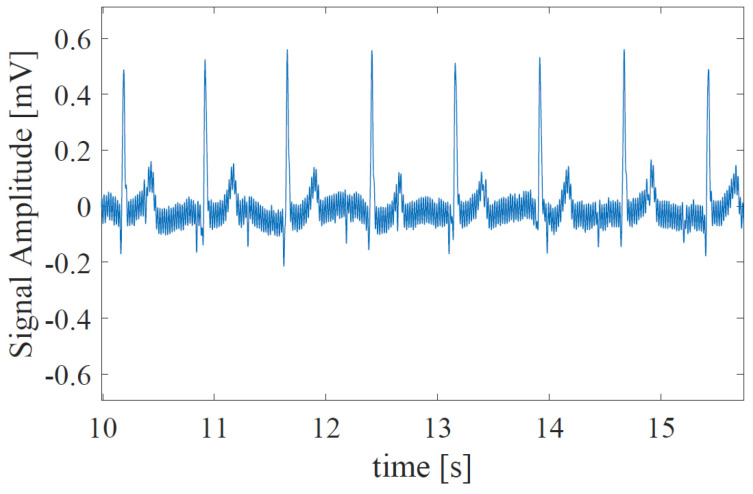
The part of AECG waveform when signal accuracy is relatively good.

**Figure 2 bioengineering-10-00048-f002:**
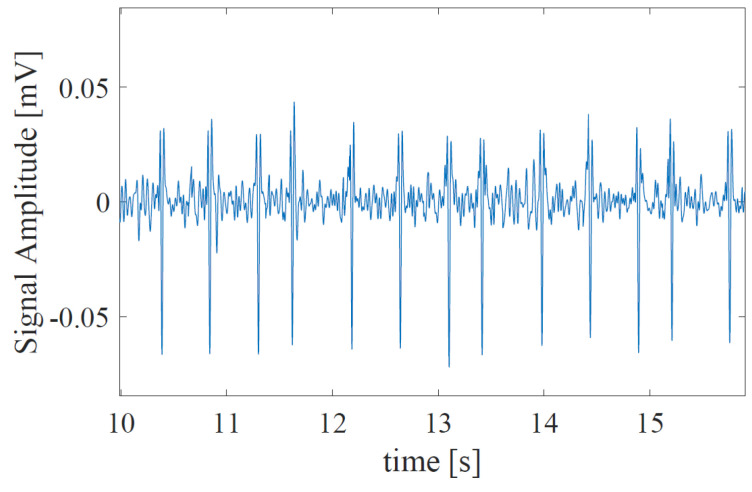
The part of the extracted FECG from AECG when the signal accuracy is relatively good.

**Figure 3 bioengineering-10-00048-f003:**
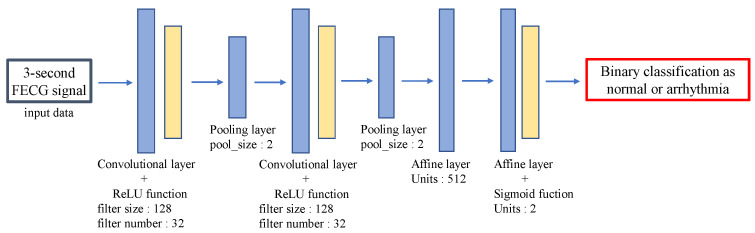
The structure of CNN model in the proposed method including layers, each parameter value, and input/output data.

**Figure 4 bioengineering-10-00048-f004:**
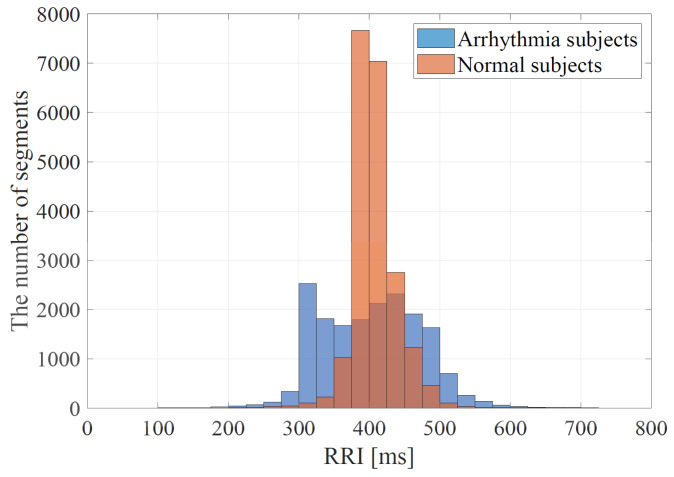
The distributions of the estimated RRIs for all the arrhythmia and normal subjects in the database used in this paper.

**Figure 5 bioengineering-10-00048-f005:**
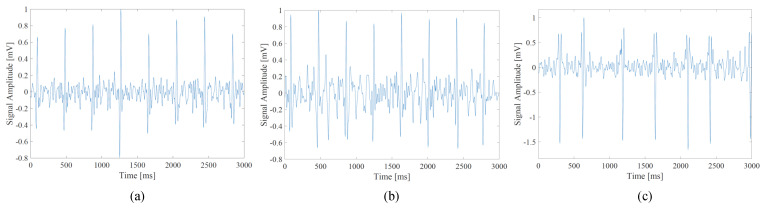
The labeled segments. (**a**) normal FECG segment; (**b**) moderate FECG segment; and (**c**) arrhythmia FECG segment.

**Figure 6 bioengineering-10-00048-f006:**
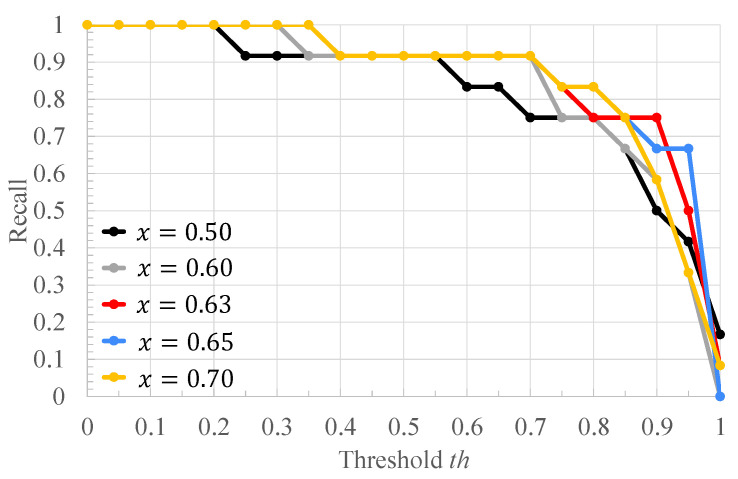
The recall of the proposed method when the threshold *x* used for RRI classification is varied.

**Figure 7 bioengineering-10-00048-f007:**
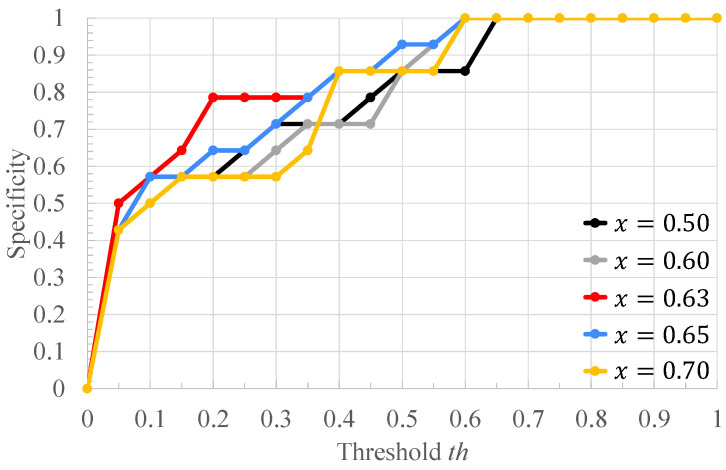
The specificity of the proposed method when the threshold *x* used for RRI classification is varied.

**Figure 8 bioengineering-10-00048-f008:**
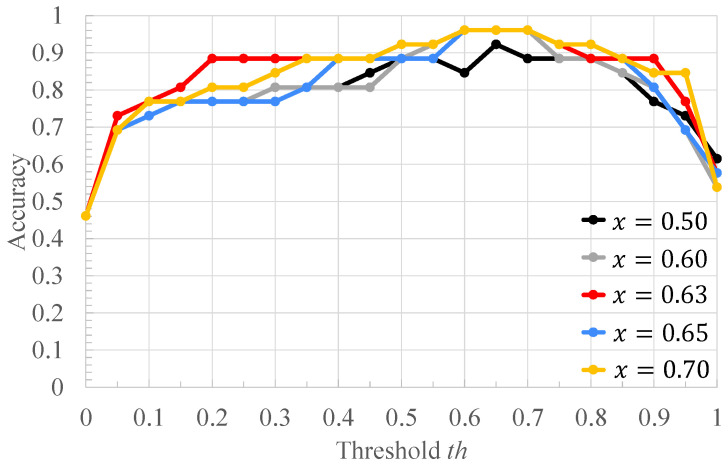
The accuracy of the proposed method when the threshold *x* used for RRI classification is varied.

**Figure 9 bioengineering-10-00048-f009:**
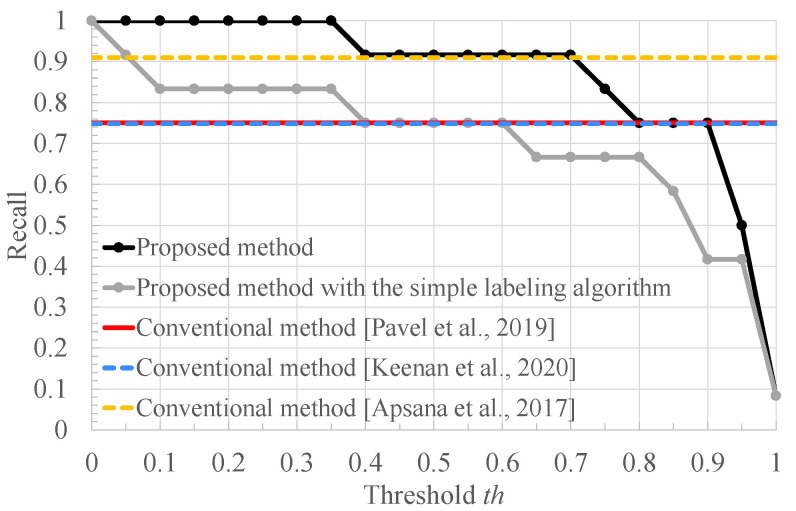
The recall of the conventional methods [8,9,10], the proposed method with the simple labeling algorithm, and the proposed method.

**Figure 10 bioengineering-10-00048-f010:**
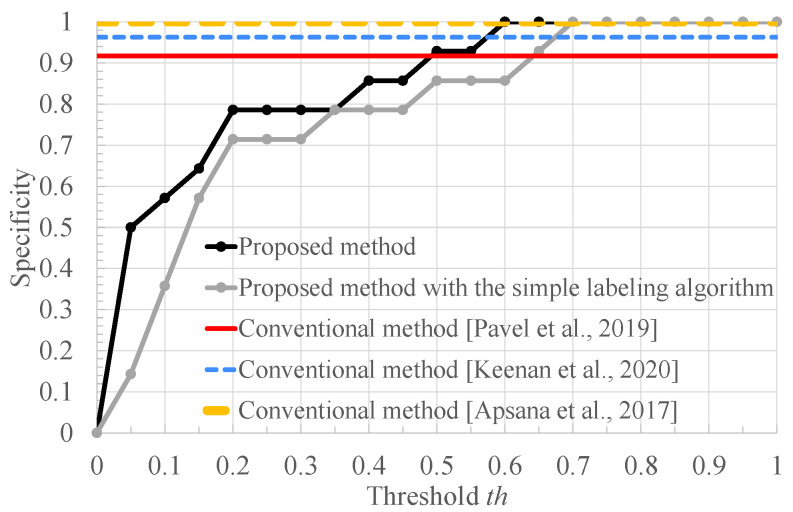
The specificity of the conventional methods [8,9,10], the proposed method with the simple labeling algorithm, and the proposed method.

**Figure 11 bioengineering-10-00048-f011:**
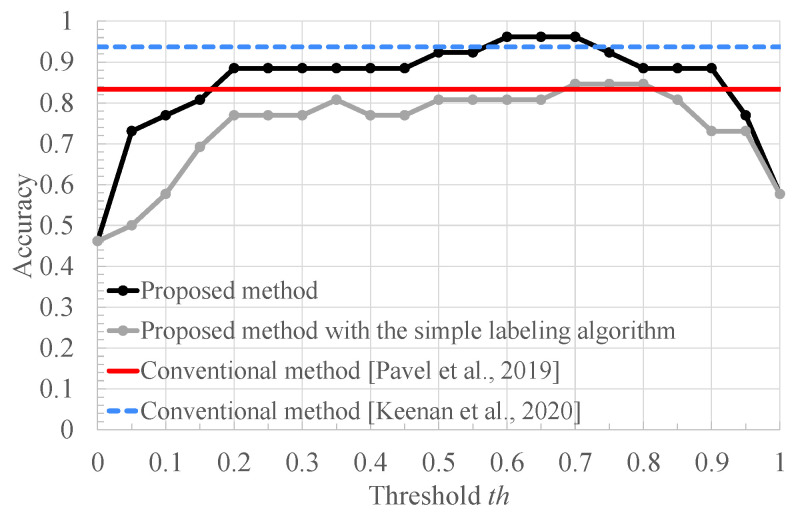
The accuracy of the conventional methods [8,9], the proposed method with the simple labeling algorithm, and the proposed method.

**Figure 12 bioengineering-10-00048-f012:**
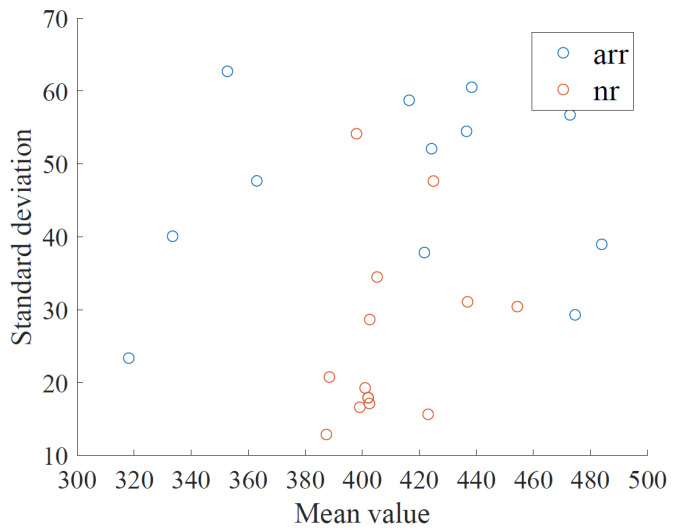
The scatter plot of the mean and standard deviation of the RRI for each of the subjects.

**Figure 13 bioengineering-10-00048-f013:**
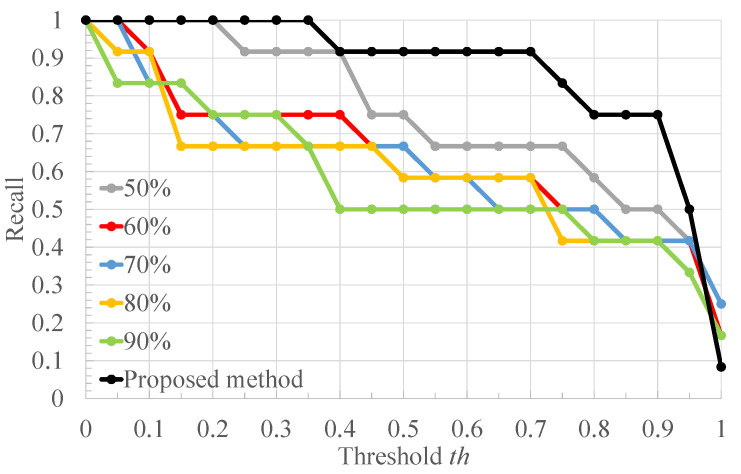
The recall of the method based on three-value classification (the percentage classified as moderate RRI = 50, 60, 70, 80, 90%) and the proposed method.

**Figure 14 bioengineering-10-00048-f014:**
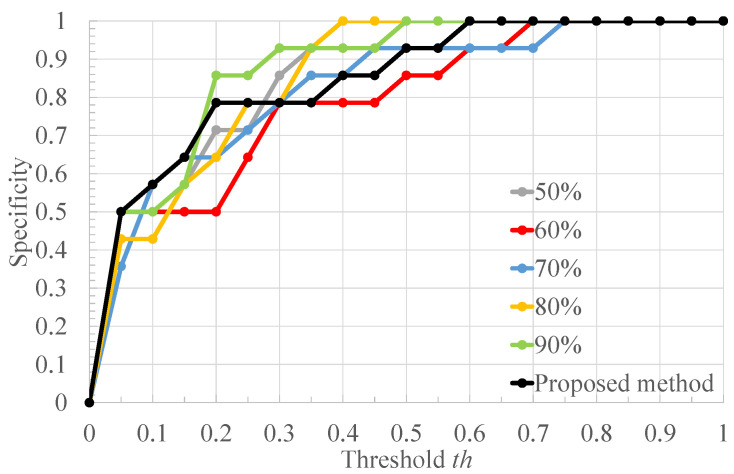
The specificity of the method based on three-value classification (the percentage classified as moderate RRI = 50, 60, 70, 80, 90%) and the proposed method.

**Figure 15 bioengineering-10-00048-f015:**
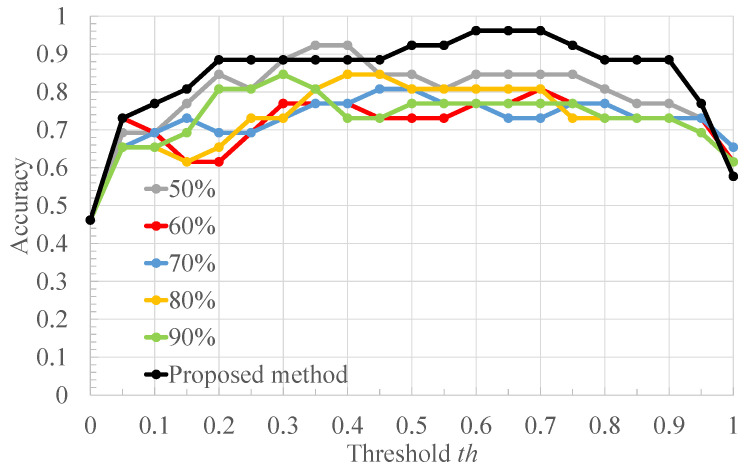
The accuracy of the method based on three-value classification (the percentage classified as moderate RRI = 50, 60, 70, 80, 90%) and the proposed method.

**Table 1 bioengineering-10-00048-t001:** Experimental specification.

Item	Value
The number of AECG	4 or 5
channels for each subject	
Sampling frequency of AECG recording	500 Hz or 1000 Hz
Measurement time	About 10 min
The number of subjects	Healthy subjects: 14
	Arrhythmia subjects: 12
Threshold *x* used to categorize RRI ranges.	0.63
Threshold th	0.00, 0.05, 0.10,..., 1.00

**Table 2 bioengineering-10-00048-t002:** The classification results of each RRI range.

RRI Range [ms]	RRIRatio *	The Classification Result				
		x=0.50	x=0.60	x=0.63	x=0.65	x=0.70
100–125	0.2000	A	A	A	A	A
125–150	0.4000	A	A	A	A	A
150–175	0.3846	A	A	A	A	A
175–200	0.0870	A	A	A	A	A
200–225	0.2000	A	A	A	A	A
225–250	0.2113	A	A	A	A	A
250–275	0.2541	A	A	A	A	A
275–300	0.1312	A	A	A	A	A
300–325	0.0427	A	A	A	A	A
325–350	0.1233	A	A	A	A	A
350–375	0.6093	M	M	A	A	A
375–400	4.2561	N	N	N	N	N
400–425	3.3120	N	N	N	N	N
425–450	1.1848	N	N	N	N	N
450–475	0.6470	M	M	M	A	A
475–500	0.2834	A	A	A	A	A
500–525	0.1530	A	A	A	A	A
525–550	0.1506	A	A	A	A	A
550–575	0.1387	A	A	A	A	A
575–600	0.2712	A	A	A	A	A
600–625	0.1389	A	A	A	A	A
625–650	0.3571	A	A	A	A	A
650–675	0.2105	A	A	A	A	A
675–700	0.2353	A	A	A	A	A
700–725	0.1111	A	A	A	A	A
725–750	0.6667	M	M	M	M	A
750–775	0.0000	A	A	A	A	A
775–800	0.0000	A	A	A	A	A
800–825	0.0000	A	A	A	A	A
825–850	0.0000	A	A	A	A	A
850–875	0.5000	M	A	A	A	A

*: The ratio of the number of healthy subjects’ RRI to the number of arrhythmia one.

**Table 3 bioengineering-10-00048-t003:** The results of labeling and classification of segments of healthy subjects.

	Labeling		Classification by CNN		The Ratio of Arrhythmia Segments
	Arrhythmia	Normal	Arrhythmia	Normal	
subject1	0	521	200	321	0.384
subject2	0	81	32	49	0.395
subject3	0	136	42	94	0.309

## Data Availability

The data used in this study are open database provided on the PhisioNet website. Available: https://physionet.org/content/nifeadb/1.0.0/ (accessed on 29 December 2022).

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
