# Peer review of "Fetal Arrhythmia Detection Based on Labeling Considering Heartbeat Interval"

_bioengineering, 2022, doi:10.3390/bioengineering10010048_

Round 1
Reviewer 1 Report
Relatively to the manuscript called “Fetal Arrhythmia Detection based on Labeling Considering Heartbeat Interval” (manuscript ID: bioengineering-2089492), I believe that excellent work is done both in terms of quality and originality. The paper is well-written and structured. The methodologies and results are sufficiently detailed and discussed. Some minor issues are found in the manuscript that should be solved to improve the article’s readability. The key issues are:
- The authors are suggested to review the abstract to report the main obtained results of the presented algorithm (also reporting numerical values).
- The authors should clarify all the acronyms at the first appearance (e.g., AECG, MECG, etc.)
- The authors should detail and enrich the Figure captions; in some cases, they are too short (e.g., Figure 1, 2,3, 4, etc.).
- The authors should specify all the hardware, firmware, and software tools used to carry out the proposed scientific work.
- The authors should review the entire manuscript to improve the English language and correct typos.
- Relatively to Table 2, the authors should better explain the structure and meaning of the table.
- The authors should review the Conclusions to summarize the main obtained results of the carried out work (also reporting numerical values).
- The authors should edit the Bibliography to comply with the journal template (for further info, see https://www.mdpi.com/journal/bioengineering/instructions).
Reviewer 2 Report
This study develops a new method to fetal arrhythmia not based on the heartbeat detection accuracy, and therefore more confident. The paper is well written and the results and methods suitably explained. However, my main concern is about the limitations, which are completely ignored by the authors. For example, does the database used in the validation include all possible sources of disturbance found in actual practice?
The Conclusion Section should include numerical values to show in a more quantitative way the merit of the new method compared to others approaches.
About the state of the art, there are some references about deep learning applied to fetal ECG proceeding which should be mentioned. The authors must discuss what is new in you study compared to these studies:
1) Boudet S, Houzé de l'Aulnoit A, Peyrodie L, Demailly R, Houzé de l'Aulnoit D. Use of Deep Learning to Detect the Maternal Heart Rate and False Signals on Fetal Heart Rate Recordings. Biosensors (Basel). 2022 Aug 27;12(9):691. doi: 10.3390/bios12090691. PMID: 36140076; PMCID: PMC9496277.
2) Mertes G, Long Y, Liu Z, Li Y, Yang Y, Clifton DA. A Deep Learning Approach for the Assessment of Signal Quality of Non-Invasive Foetal Electrocardiography. Sensors (Basel). 2022 Apr 26;22(9):3303. doi: 10.3390/s22093303. PMID: 35591004; PMCID: PMC9103336.
Minor details: Line 104 and 106: AECG and MECG have to be defined.
Round 2
Reviewer 2 Report
The authors have adequately addressed my comments.